DOI: 10.1038/s41467-018-06023-5　　**OPEN**

# The complex underpinnings of genetic background effects

Martin N. Mullis[1], Takeshi Matsui[1], Rachel Schell[1], Ryan Foree[1] & Ian M. Ehrenreich[1]

Genetic interactions between mutations and standing polymorphisms can cause mutations to show distinct phenotypic effects in different individuals. To characterize the genetic architecture of these so-called background effects, we genotype 1411 wild-type and mutant yeast cross progeny and measure their growth in 10 environments. Using these data, we map 1086 interactions between segregating loci and 7 different gene knockouts. Each knockout exhibits between 73 and 543 interactions, with 89% of all interactions involving higher-order epistasis between a knockout and multiple loci. Identified loci interact with as few as one knockout and as many as all seven knockouts. In mutants, loci interacting with fewer and more knockouts tend to show enhanced and reduced phenotypic effects, respectively. Cross–environment analysis reveals that most interactions between the knockouts and segregating loci also involve the environment. These results illustrate the complicated interactions between mutations, standing polymorphisms, and the environment that cause background effects.

---

[1] Molecular and Computational Biology Section, Department of Biological Sciences, University of Southern California, Los Angeles, CA 90089-2910, USA. These authors contributed equally: Martin N. Mullis, Takeshi Matsui, Rachel Schell. Correspondence and requests for materials should be addressed to M.N.M. (email: mmullis@usc.edu) or to T.M. (email: tmatsui@usc.edu) or to I.M.E. (email: ian.ehrenreich@usc.edu)

Background effects occur when the same spontaneous or induced mutations show different phenotypic effects across genetically distinct individuals[1–7]. Countless examples of background effects have been described across species and traits[1,2], collectively suggesting that this phenomenon is common in biological systems and plays a significant role in many phenotypes. For example, alleles that show background effects contribute to a wide range of hereditary disorders, including, but not limited to, colorectal cancer, hypertension, and phenylketonuria[8]. Background effects may also impact other disorders that frequently involve de novo mutations, such as autism[9], congenital heart disease[10], and schizophrenia[11]. Additionally, it has been proposed that background effects can shape the potential trajectories of evolutionary adaptation[12,13], influence the emergence of novel traits[7], and help maintain deleterious genetic variation within populations[14].

Despite the importance of background effects to biology and medicine, understanding of their causal genetic mechanisms remains limited. Although superficially background effects are known to arise due to genetic interactions (or "epistasis") between mutations and standing polymorphisms[15–20], only recently have studies begun to provide deeper insights into the architecture of epistasis underlying background effects. These papers indicate that background effects often involve multiple polymorphisms that interact not only with a mutation but also with each other[4–7,21–26] and the environment[6]. This work also suggests that background effects are caused by a mixture of loci that show enhanced and reduced phenotypic effects in mutants relative to wild-type individuals[1,5–7,18,23–25,27–32]. Together, these previous reports imply that the phenotypic effect of a mutation in a given genetic background can depend on an individual's genotype at a potentially large number of loci that interact in complicated, highly contextual ways. However, this point is difficult to explicitly show because doing so requires systematically mapping the interactions between mutations, polymorphisms, and environment that give rise to background effects.

In this paper, we perform a detailed genetic characterization of a number of background effects across multiple environments. Previous work in yeast, as well as other model species, has established that mutations in chromatin regulation and transcription often show background effects[5–7,21,28,29,33]. We extend this past work by knocking out seven different chromatin regulators in a cross of the BY4716 (BY) and 322134S (3S) strains of *Saccharomyces cerevisiae*. We generate and genotype 1411 wild-type and knockout segregants, measure the growth of these individuals in 10 environments, and perform linkage mapping with these data. In total, we identify 1086 interactions between the knockouts and segregating loci. These interactions allow us to obtain novel, detailed insights into the genetic architecture of background effects across different mutations and environments.

## Results

### Preliminary screen
When a mutation that exhibits background effects is introduced into a population, the phenotypic variance among individuals will often change[24,25,27,30]. Here, we attempted to identify mutations that induce such changes in phenotypic variance. Specifically, we screened 47 complete gene knockouts of histones, histone-modifying enzymes, chromatin remodelers, and other chromatin-associated genes for impacts on phenotypic variance in segregants from a cross of the BY and 3S strains of budding yeast (Supplementary Figs. 1 and 2; Supplementary Table 1; Methods). To do this, we generated BY/3S diploid hemizygotes, sporulated these hemizygotes to obtain haploid knockout segregants, and then quantitatively phenotyped these BY×3S knockout segregants for growth on rich medium

containing ethanol, an environment in which we previously found background effects that influence yeast colony morphology[5–7,34]. For each panel of segregants, three biological replicate end-point colony growth assays were performed and averaged. We then tested whether the knockout segregants exhibited significantly higher phenotypic variance than wild-type segregants using Levene's test (Supplementary Table 2). This analysis implicated *CTK1* (a kinase that regulates RNA polymerase II), *ESA1* and *GCN5* (two histone acetyltransferases), *HOS3* and *RPD3* (two histone deacetylases), *HTB1* (a copy of histone H2B), and *INO80* (a chromatin remodeler) as knockouts that show background effects in the BY×3S cross (Supplementary Fig. 1; Supplementary Note 1).

### Mapping of mutation-independent and mutation-responsive effects
To map loci that interact with the seven knockouts identified in the screen, we genotyped 1411 segregants in total. This included 164 wild-type, 210 *ctk1Δ*, 122 *esa1Δ*, 215 *gcn5Δ*, 220 *hos3Δ*, 177 *htb1Δ*, 141 *ino80Δ*, and 162 *rpd3Δ* segregants (Supplementary Fig. 3; Supplementary Tables 3 through 5; Methods). These genotyped segregants were phenotyped for growth in 10 diverse environments using replicated end-point colony growth assays (Supplementary Fig. 4; Supplementary Table 6; Methods). We note that, despite causing increased phenotypic variance in ethanol, the knockouts induced a broad range of phenotypic responses in other environments (Supplementary Fig. 4).

As described in detail in Methods, genome-wide linkage mapping scans were conducted within each environment (Supplementary Data 1 and 2). To maximize statistical power, we analyzed the 1411 segregants jointly using a fixed-effects linear model that accounted for genetic background. We identified individual loci, as well as two- and three-way genetic interactions among loci, that exhibited the same phenotypic effect across the wild-type and knockout backgrounds (hereafter "mutation-independent" effects). We also conducted scans for individual loci, as well as two- and three-way genetic interactions among loci, that exhibited different phenotypic effects in at least one knockout background relative to the wild-type background (hereafter "mutation-responsive" effects). Post hoc tests were used to associate mutation-responsive effects with specific knockouts. Mutation-responsive one-, two-, and three-locus effects can alternatively be viewed as two-, three-, and four-way interactions where one of the involved genetic factors is a knockout. However, to avoid confusion throughout the paper, we do not count the knockouts as genetic factors. Instead, we classify each genetic effect as mutation-independent or –responsive and report how many loci it involves. Representative examples of mutation-responsive effects are shown in Fig. 1.

In total, we detected 1211 genetic effects across the 10 environments (Supplementary Figs. 5 and 6; Supplementary Data 3; Supplementary Tables 7 through 10; Supplementary Note 2 and 3). One hundred and twenty five (10%) of these genetic effects were mutation-independent while 1086 (90%) were mutation-responsive (Fig. 2a). On average, we identified 121 genetic effects per environment, 109 of which were mutation-responsive. However, the number of detected genetic effects varied significantly across the environments, from 15 in room temperature to 359 in ethanol. Despite this variability, in every environment, ≥47% of the identified genetic effects were mutation-responsive. This suggests that, regardless of environment, most genetic effects in the cross were responsive to the knockouts. Additionally, the seven knockouts exhibited major differences in their numbers of mutation-responsive effects. Between 73 and 118 mutation-responsive effects were found for

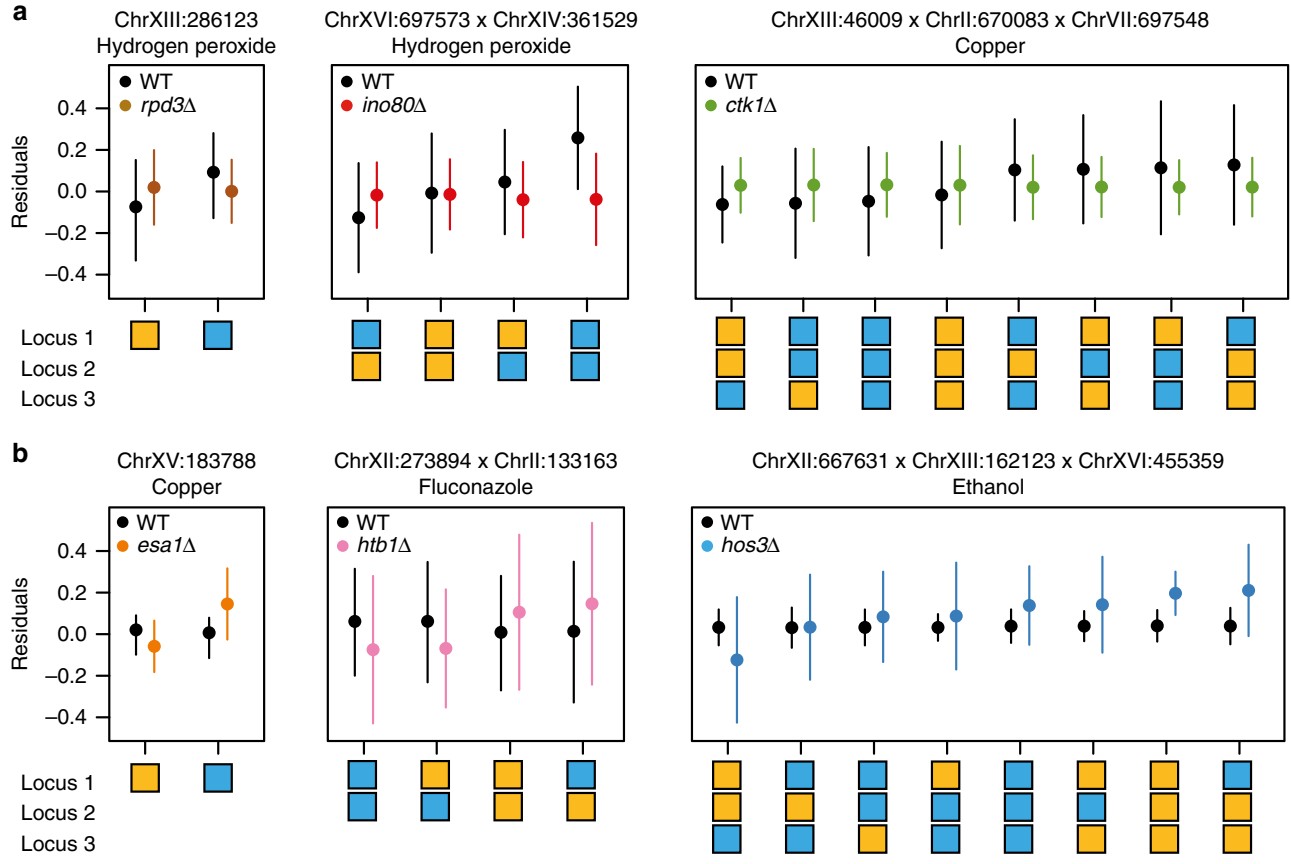

**Fig. 1** Examples of mutation-responsive genetic effects. **a** shows representative examples of one-, two-, and three-locus mutation-responsive effects with larger phenotypic effects in wild-type segregants than mutants. In contrast, **b** shows representative examples of one-, two-, and three-locus mutation-responsive effects with larger phenotypic effects in mutants than wild-type segregants. Means depicted along the y axis show residuals from a fixed-effects linear model that includes the mutation-independent effect of each involved locus, as well as any possible lower-order mutation-independent and mutation-responsive effects. The different genotype classes are plotted below the x axis. Blue and orange boxes correspond to the BY and 3S alleles of a locus, respectively. Error bars represent one standard deviation from the mean

the *CTK1*, *ESA1*, *GCN5*, *HTB1*, *INO80*, and *RPD3* knockouts (Fig. 2b). In contrast, the *HOS3* knockout had 543 mutation-responsive effects (Fig. 2b).

**Higher-order epistasis influences response to mutations**. While only 29% (36 of 125) of the mutation-independent effects involved multiple loci, this proportion was more than tripled (89%; 965 of 1086) among the mutation-responsive effects (Fig. 2a). Simulations indicate that our statistical power to detect mutation-responsive loci was appreciably higher for single locus effects than for multiple locus effects, suggesting that our results may underestimate the importance of higher-order epistasis to background effects (Supplementary Fig. 7). To better assess how loci involved in the identified higher-order interactions contribute to background effects, we partitioned the individual and joint contributions of involved loci to mutation-responsive phenotypic variance (Supplementary Data 4 and 5; Methods). For mutation-responsive two-locus effects, on average, 78% of the mutation-responsive phenotypic variance was attributed to the higher-order interaction between the knockout and both loci (Fig. 3a). Likewise, among mutation-responsive three-locus effects, on average, 58% of the mutation-responsive phenotypic variance was explained by the higher-order interaction of the knockout and the three loci (Fig. 3b). Thus, most mutation-responsive effects involve multiple loci that contribute to background effects predominantly through their higher-order interactions with each other and a mutation, rather than through their individual interactions with a mutation.

**Environment plays a strong role in background effects**. The role of the environment in background effects has yet to be fully characterized. Although our group previously showed that the genetic architecture of background effects can significantly change across environments[6], this past work focused on only a modest number of segregating loci and environments. To more generally assess how the environment influences the genetic architecture of background effects, we determined whether the 1086 mutation-responsive effects impacted phenotype in environments outside the ones in which they were originally detected. This analysis was performed using statistical thresholds that were more liberal than those employed in our initial genetic mapping (Methods). In all, 29% (311) of the mutation-responsive effects were detectable in additional environments, with this proportion varying between 7% and 65% across the 10 environments. (Fig. 4). Of these mutation-responsive effects, 64% (200) were identified in only one additional environment, 28% (85) were found in two additional environments, and just 8% (26) were detected in ≥3 environments. Given the limited resolution of the data, it is possible that some of the mutation-responsive effects that were detected in multiple environments in fact represent distinct, closely linked loci that act in different environments. Such linkage would lead us to overestimate how often mutation-responsive effects contribute to background effects in

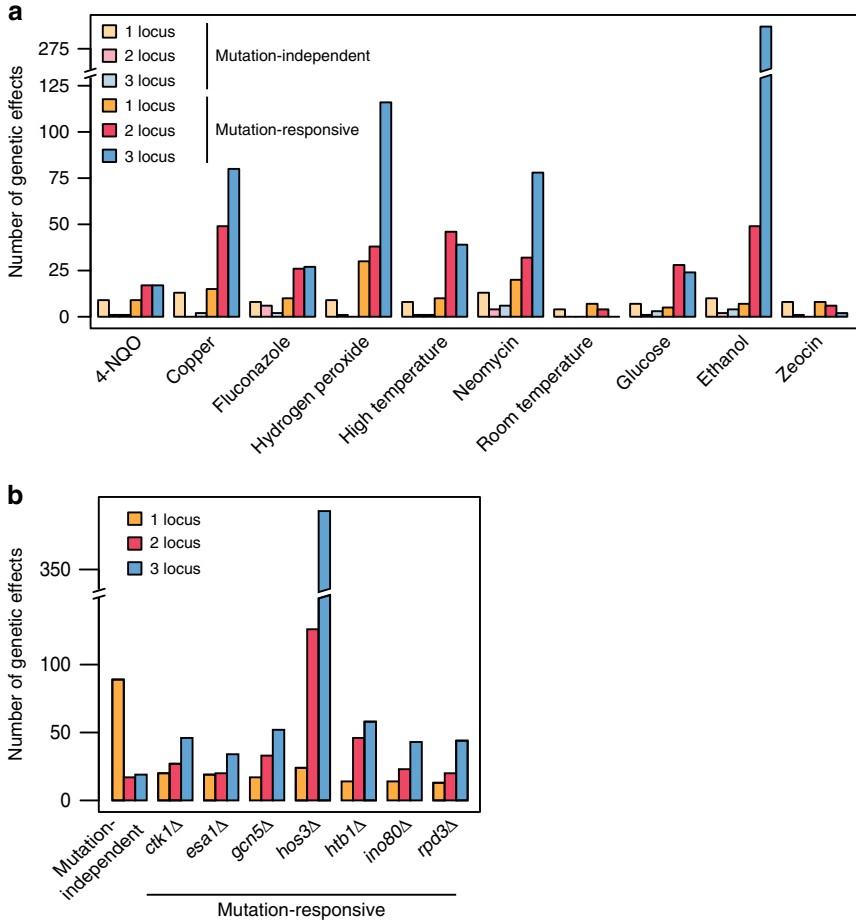

**Fig. 2** Most mutation-responsive genetic effects involve multiple loci. In **a**, the number of mutation-independent and mutation-responsive genetic effects detected in each environment are shown. In **b**, the aggregate numbers of mutation-responsive effects found for each knockout across the 10 environments are provided

different environments, further suggesting that most mutation-responsive effects act in a limited number of environments. These findings support the idea that background effects are caused by complex interactions between not only mutations and polymorphisms but also the environment.

**Interactions between loci and different knockouts.** We next looked at how the same mutation-responsive effects interact with different knockouts. Based on involvement of the same loci, the 1086 mutation-responsive effects were collapsed into 594 distinct mutation-responsive effects that showed epistasis with at least one knockout (Methods). In all, 65% of these mutation-responsive effects were found in only one knockout background, while 35% were identified in ≥2 knockout backgrounds (Supplementary Fig. 8). Also, 97% of the mutation-responsive effects that interacted with only one knockout were *HOS3*-responsive and these effects represented 69% of the total interactions detected in a *hos3Δ* background (Fig. 5a). In contrast, nearly all (between 95% and 100%) of the *CTK1-*, *ESA1-*, *GCN5-*, *HTB1-*, *INO80-*, and *RPD3*-responsive effects were detected in multiple backgrounds (Fig. 5a). Although the mutation-responsive effects exhibited a broad, continuous range of responses to the knockouts (Fig. 5b), they could be partitioned into two qualitative classes—enhanced and reduced—based on whether they explained more or less phenotypic variance in mutants than in wild-type segregants, respectively. The distinct mutation-responsive effects exhibited a strong relationship between their number of interacting knock-outs and how they were classified. Mutation-responsive effects

that interacted with fewer than three knockouts predominantly were in the enhanced class, while mutation-responsive effects that interacted with four or more knockouts typically were in the reduced class ($\chi^2 = 709.37$, d.f. = 6, $p = 5.81 \times 10^{-150}$; Fig. 5c; Supplementary Data 6). These results illustrate how background effects are caused by a mixture of loci that respond specifically to mutations in particular genes and loci that respond more generically to mutations in different genes, with the relative contribution of these two classes of loci varying significantly across mutations. Our findings also suggest that how loci respond to mutations in a particular gene is related to the degree to which they interact with mutations in other genes.

**Genetic basis of induced changes in phenotypic variance.** Lastly, we looked at the extent to which the identified mutation-responsive effects in aggregate related to differences in phenotypic variance between the knockout and wild-type versions of the BY×3S cross across the 10 environments (Supplementary Table 11). This was important because many studies (e.g., refs. [23–25,29,30,35]) have described how perturbing certain genes can alter the phenotypic variance within a population, but the genetic underpinnings of this phenomenon have not been fully determined. Among the 70 different combinations of the 7 knockouts and 10 environments, we found a highly significant relationship between differences in the numbers of mutation-responsive effects with reduced and enhanced phenotypic effects and knockout-induced changes in phenotypic variance (Fig. 6; Spearman's $\rho = 0.84$, $p = 4.33 \times 10^{-20}$). No such relationship was seen when we looked at the mean phenotypic

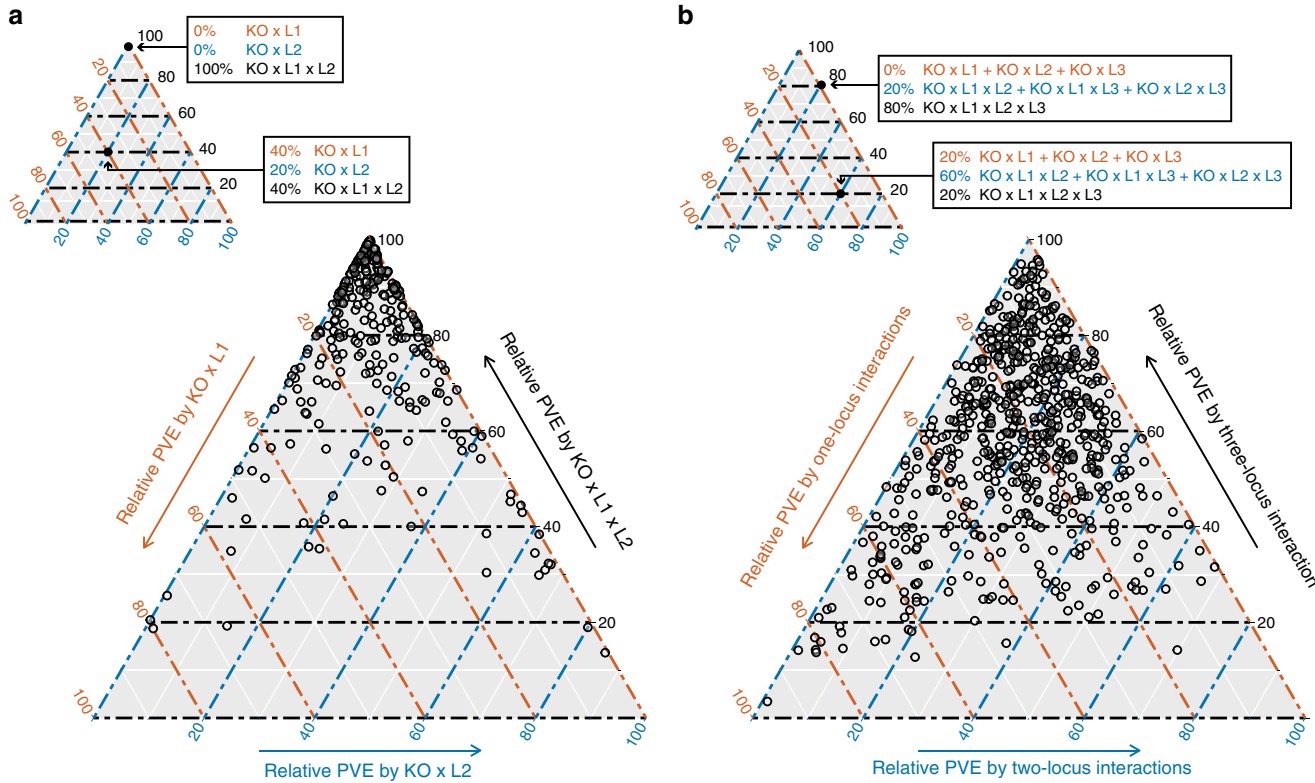

**Fig. 3** Higher-order epistasis among knockouts and multiple loci is an important contributor to background effects. In **a**, for each mutation-responsive two-locus effect, we partitioned the individual and joint contributions of the two loci. L1 and L2 refer to the involved loci, while KO denotes the relevant knockout. We determined the relative phenotypic variance explained (PVE) by interactions between a knockout and each individual locus (i.e., KO × L1 and KO × L2) and higher-order epistasis involving a knockout and the two loci (i.e., KO × L1 × L2). Similarly, in **b**, for each mutation-responsive three-locus effect, we determined the relative PVE for all possible mutation-responsive one-, two-, and three-locus effects involving the participating loci. In both **a** and **b**, relative PVE values were calculated using sum of squares obtained from ANOVA tables, as described in Methods. Mutation-responsive effects that interact with multiple knockouts are shown multiple times, once for each relevant knockout

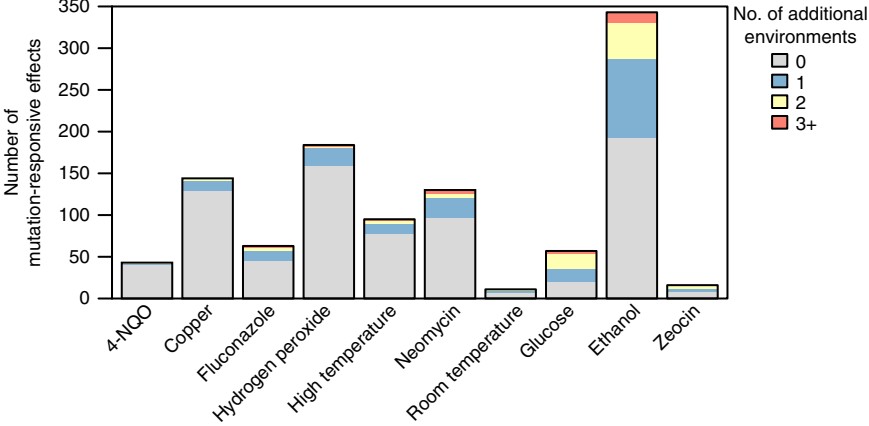

**Fig. 4** Analysis of mutation-responsive effects across environments. The height of each stacked bar indicates the number of mutation-responsive effects that were detected in a given environment. The bars are color-coded according to the number of additional environments in which these mutation-responsive effects could be detected when liberal statistical thresholds were employed (Methods)

changes induced by the mutations (Supplementary Fig. 9). To control for potential biases in our analyses that might arise from allele frequency differences among the backgrounds (Supplementary Fig. 3), we performed the same analysis on each knockout background individually using data from the 10 environments (Supplementary Table 11). When we did this, we found that all seven knockout backgrounds exhibited nominally significant correlations between observed changes in phenotypic variance and

detected mutation-responsive effects (Spearman's $\rho > 0.71$, $p < 0.02$; Supplementary Fig. 10). Permutations indicate that the probability of observing this result by chance is low ($p < 10^{-5}$). Thus, these findings are consistent with the knockout-induced changes in phenotypic variance resulting from a large number of epistatic loci with small phenotypic effects (Supplementary Fig. 11; Supplementary Table 11). In summary, our results not only provide valuable insights into the genetic architecture of background effects but also

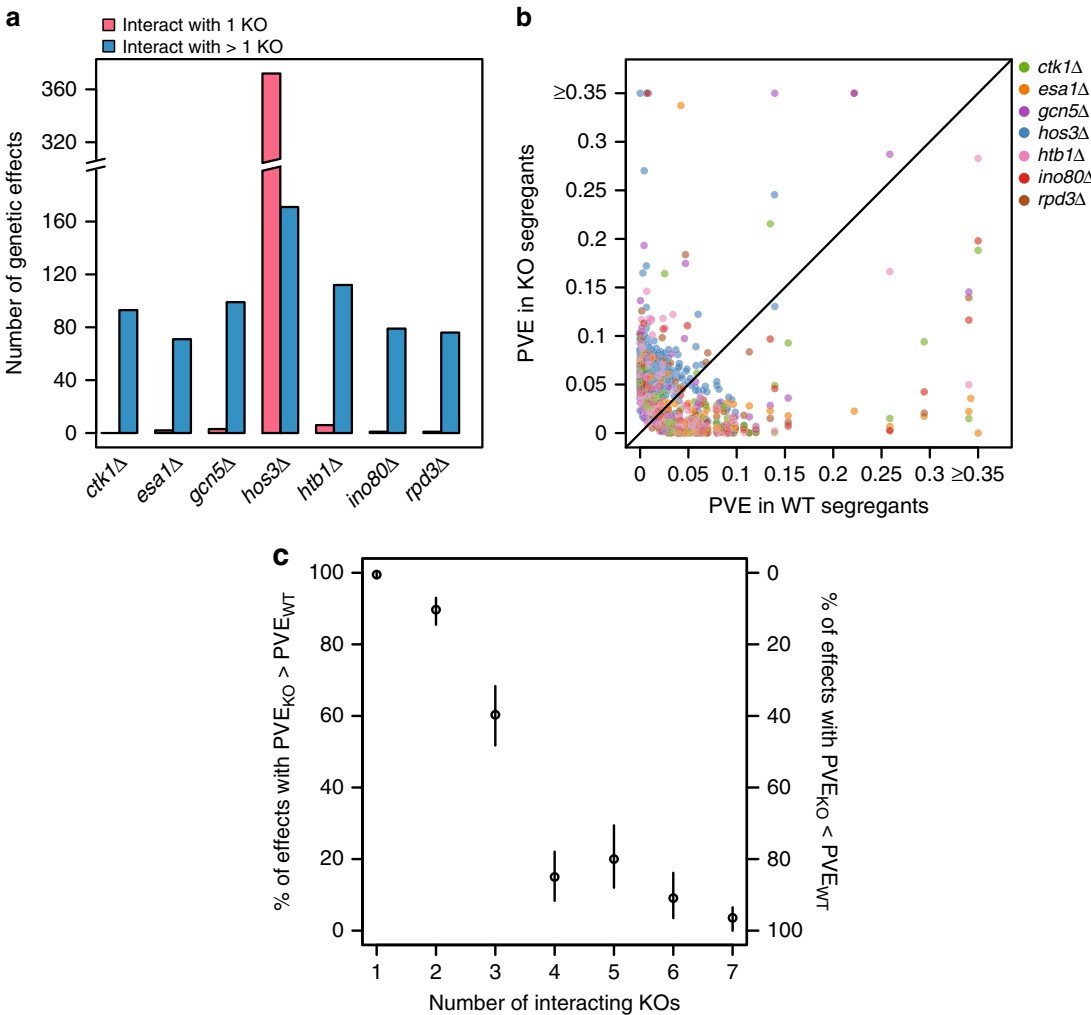

**Fig. 5** Analysis of mutation-responsive effects across knockout backgrounds. In **a**, the number of mutation-responsive effects that interacted with only one knockout (pink) or interacted with multiple knockouts (blue) are shown for each knockout. In **b**, the phenotypic variance explained (PVE) for each mutation-responsive effect is shown in the relevant knockout (KO) segregants, as well as in the wild-type (WT) segregants. The PVE for each mutation-responsive effect was determined using fixed-effects linear models fit within each individual background (Methods). Mutation-responsive effects are color-coded by the knockout population in which they were identified. In **c**, the percentage of mutation-responsive effects that showed larger phenotypic effects in mutants than in wild-type segregants (y axis, left side) and mutation-responsive effects that showed larger phenotypic effects in wild-type segregants than in mutants (y axis, right side) is depicted. These values are plotted as a function of the number of knockouts that interact with a given mutation-responsive effect. Error bars represent 95% bootstrap confidence intervals (Methods)

illustrate how interactions between mutations, segregating loci, and the environment can influence a population's phenotypic variance.

## Discussion
Most prior studies of background effects have described specific examples without identifying the contributing loci. Here we used a screen of 47 different chromatin regulators to identify 7 knockout mutations that exhibit strong background effects in a yeast cross. We then generated and phenotyped a panel of 1411 mutant and wild-type segregants. Using these data, we detected 1086 genetic interactions that involve the 7 knockouts and loci that segregate in the cross. To better understand the genetic architecture of background effects, we comprehensively examined how the identified loci interact not only with the knockouts but also with each other and the environment. Our results confirm important points about the genetic architecture of background effects that to date have been suggested but not conclusively proven. Namely, background effects can be highly polygenic, with many, if not most, loci contributing through higher-order genetic

interactions that involve a mutation and multiple loci. These loci can respond to mutations in different ways, such as by exhibiting enhanced and reduced phenotypic effects in mutants relative to wild-type individuals. Moreover, most of these interactions between mutations and segregating loci also involve the environment. Altogether, these findings shed light on the complex genetic and genotype–environment interactions that give rise to background effects.

Our work also illustrates how the genetic architecture of background effects varies significantly across different mutated genes. In our study, response to six of the seven knockouts was mediated almost exclusively by loci that respond to mutations in different genes and predominantly exhibit reduced effects in mutants relative to wild-type segregants. Given that some of the examined chromatin regulators have counteracting or unrelated biochemical activities[36,37], we propose that loci detected in multiple knockout backgrounds respond generically to perturbations of cell state or fitness, rather than to any specific biochemical process. In contrast, response to *HOS3* knockout was largely

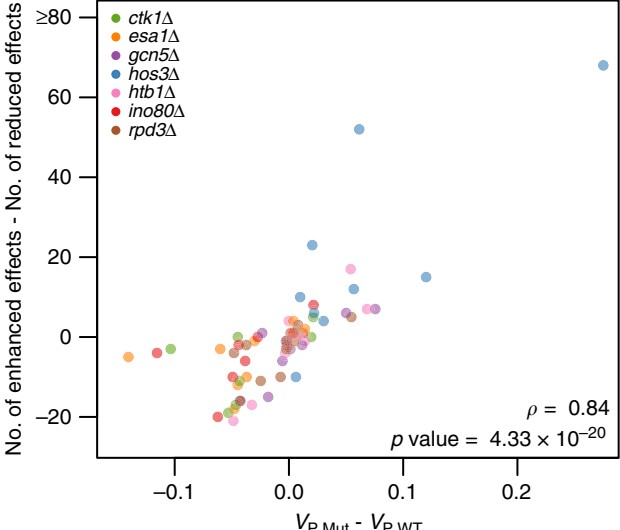

**Fig. 6** Mutation-responsive effects underlie differences in phenotypic variance between knockout and wild-type backgrounds across environments. Each point's position on the x axis represents the difference in phenotypic variance between a knockout background of the cross ($V_{P.Mut}$) and the wild-type background of the cross ($V_{P.WT}$) in a single environment. The y axis shows the difference in the number of mutation-responsive effects with enhanced and reduced phenotypic effects. In this paper, we classified mutation-responsive effects as enhanced or reduced based on whether they explained more or less phenotypic variance in mutants relative to wild-type segregants, respectively. Spearman's $\rho$ and its associated p value are provided on the plot. Colors denote different knockout backgrounds

mediated by loci that were not detected when the other genes were compromised. Why so many loci responded specifically to perturbation of *HOS3* is difficult to infer from current understanding of Hos3's biochemical activities. Although Hos3 can deacetylate all four of the core histones[38] and influence chromatin regulation in certain genomic regions[39], it also plays roles in cell cycle[40] and nuclear pore regulation[41]. Thus further work is needed to characterize *HOS3* and its extensive epistasis with polymorphisms in the BY×3S cross.

In addition to advancing understanding of background effects, our results may also have more general implications for the genetic architecture of complex traits. Many phenotypes, including common disorders like autism[9] and schizophrenia[11], are influenced by loss-of-function mutations that occur de novo or persist within populations at low frequencies. We have shown that these mutations can significantly change the phenotypic effects of many polymorphisms within a population by altering how these polymorphisms interact with each other and the environment. Although these complicated interactions between mutations, standing polymorphisms, and the environment are often ignored in genetics research, our study suggests that they in fact play a major role in determining the relationship between genotype and phenotype.

## Methods

**Generation of different BY×3S knockout backgrounds**. All BY×3S segregants described in this paper were generated using the synthetic genetic array marker system, which makes it possible to obtain *MATa* haploids by digesting tetrads and selecting for spores on minimal medium lacking histidine and containing canavanine[42] (Supplementary Fig. 1). We first constructed a BY/3S diploid by mating a BY *MATa can1Δ::STE2pr-SpHIS5 his3Δ* strain to a 3S *MATα ho::HphMX his3Δ::NatMX* strain. This diploid served as the progenitor for the wild-type segregants. Hemizygous complete gene deletions were engineered into this wild-type BY/3S diploid to produce the progenitors of the knockout segregants. Genes were deleted

using transformation with PCR products that were comprised of (in the following order) 60 bp of genomic sequence immediately upstream of the targeted gene, *KanMX*, and 60 bp of genomic sequence immediately downstream of the targeted gene. Lithium acetate transformation was employed[43]. To obtain a given knockout, transformants were selected on rich medium containing G418, ClonNAT, and Hygromycin B, and PCR was then used to check transformants for correct integration of the *KanMX* cassette. These PCRs were conducted with primer pairs where one primer was located within *KanMX* and the other primer was located adjacent to the expected site of integration. PCR products were Sanger sequenced. Primers used in these checks are reported in Supplementary Table 18. Wild-type and hemizygous knockout diploids were sporulated using standard techniques. Low-density random spore plating (around 100 colonies per plate) was then used to obtain haploid BY×3S segregants from each wild-type and knockout background of the cross. Wild-type segregants were isolated directly from *MATa* selection plates, while knockout segregants were first replica plated from *MATa* selection plates onto G418 plates, which selected for the gene deletions.

**Genotyping of segregants**. Segregants were genotyped using low-coverage whole-genome sequencing. A sequencing library was prepared for each segregant using the Illumina Nextera Kit and custom barcoded adapters. Libraries from different segregants were pooled in equimolar fractions and these multiplex pools were size selected using the Qiagen Gel Extraction Kit. Multiplexed samples were sequenced by BGI on an Illumina HiSeq 2500 using 100 bp × 100 bp paired-end reads. For each segregant, reads were mapped against the S288c genome (version S288C_reference_sequence_R64-2-1_20150113.fsa from https://www.yeastgenome.org) using BWA version 0.7.7-r44[44]. Pileup files were then produced with SAMTOOLS version 0.1.19-44428 cd[45]. BWA and SAMTOOLS were run with their default settings. Base calls and coverages were obtained from the pileup files for 36,756 previously identified high-confidence single-nucleotide polymorphisms (SNPs) that segregate in the cross[7]. Individuals who showed evidence of being aneuploid, diploid, or contaminated based on unusual patterns of coverage or heterozygosity were excluded from further analysis. We also used the data to confirm the presence of *KanMX* at the gene that had been knocked out. Individuals with an average per site coverage <1.5× were removed from the dataset. A vector containing the fraction of 3S calls at each SNP was generated and used to make initial genotype calls with sites above and below 0.5 classified as 3S and BY, respectively. This vector of initial genotype calls was then corrected with a Hidden Markov Model (HMM), implemented using the HMM package version 1.0 in R[46]. We used the following transition and emission probability matrices: transProbs = matrix(c(.9999,.0001,.0001,.9999),2) and emissionProbs = matrix(c(.0.25,0.75,0.75,0.25),2). We examined the HMM-corrected genotype calls for adjacent SNPs that lacked recombination in the segregants. In such instances, a single SNP was chosen to serve as the representative for the entire set of adjacent SNPs that lacked recombination. This reduced the number of markers used in subsequent analyses from 36,756 to 8311.

**Phenotyping of segregants**. Prior to phenotyping, segregants were always inoculated from freezer stocks into YPD broth containing 1% yeast extract (BD Product #: 212750), 2% peptone (BD 211677), and 2% dextrose (BD Product #:15530). After these cultures had reached stationary phase, they were pinned onto and outgrown on plates containing 2% agar (BD 214050). Unless specified, these plates were made with YPD and incubated at 30 °C for 2 days. However, some of the environments required adding a chemical compound to the YPD plates or changing the temperature or carbon source. In addition to YPD at 30 °C, we measured growth in the following environments: 21 °C, 42 °C, 2% ethanol (Koptec A06141602W), 250 ng/mL 4-nitroquinoline 1-oxide ("4NQO") (TCI N0250), 9 mM copper sulfate (Sigma 209198), 50 mg/mL fluconazole (TCI F0677), 260 mM hydrogen peroxide (EMD Millipore HX0640-5), 7 mg/mL neomycin sulfate (Gibco 21810-031), and 5 mg/mL zeocin (Invivogen ant-zn-1). For 4-NQO, copper sulfate, fluconazole, hydrogen peroxide, neomycin sulfate, and zeocin, the doses used for phenotyping were chosen based on preliminary experiments across a broader range of concentrations (Supplementary Table 6). Growth assays were conducted in triplicate using a randomized block design to account for positional effects on the plates. Four BY controls were included on each plate. Plates were imaged using the BioRAD Gel Doc XR+ Molecular Imager. Each image was 11.4 × 8.52 cm² (width × length) and imaged under white epi-illumination with an exposure time of 0.5 s. Images were exported as Tiff files with a resolution of 600 dpi. As in ref. [47], image analysis was conducted in the ImageJ software, with pixel intensity for each colony calculated using the Plate Analysis JRU v1 plugin (http://research.stowers.org/imagejplugins/index.html). The growth of each segregant on each plate was computed by dividing the segregant's total pixel intensity by the mean pixel intensity of the average of BY controls from the same plate. The replicates for a segregant within an environment were then averaged and used as that individual's phenotype in subsequent analyses.

**Scans for one-locus effects**. All genetic mapping was conducted within each environment using fixed-effects linear models applied to the complete set of 1411 wild-type and knockout segregants. To ensure that mean differences in growth among the eight backgrounds were always controlled for during mapping, we

included a *background* term in our models. Throughout the paper, we refer to loci or combinations of loci that statistically interact or do not statistically interact with the *background* term as mutation-responsive and mutation-independent, respectively. Genetic mapping was performed in R using the lm() function, with the $p$ values for relevant terms obtained from tables generated using the summary() function.

We first identified individual loci that show mutation-independent or mutation-responsive phenotypic effects using forward regression. To detect mutation-independent loci, genome-wide scans were conducted with the model *phenotype ~ background + locus + error*. Significant loci identified in this first iteration were then used as covariates in the next iteration, i.e., *phenotype ~ background + known_locus1 … known_locusN + locus + error*, where the *known_locus* terms corresponded to each of the loci that had already been identified in a given environment. To determine significance, 1000 permutations were conducted at each iteration of the forward regression, with the correspondence between genotypes and phenotypes randomly shuffled each time. Among the minimum $p$ values obtained in the permutations, the fifth quantile was identified and used as the threshold for determining significant loci. This process was iterated until no additional loci could be detected for each environment.

To identify mutation-responsive one-locus effects, we employed the same procedure described in the preceding paragraph, except the model *phenotype ~ background + locus + background:locus + error* was used. Here the significance of the *background:locus* interaction term was tested, again with significance determined by permutations as described in the preceding paragraph. The *locus* term was included in the model to ensure that phenotypic variance explained by mutation-independent effects did not load onto the mutation-dependent effects. For each locus with significant *background:locus* terms, we included both an additive and *background* interaction term in the subsequent iterations of the forward regression: i.e., *phenotype ~ background + known_locus1 … known_locusN + locus + background:known_locus1 … background:known_locusN + background: locus + error*. The *known_locus* terms were included in these forward regression models to ensure that variance due to the mutation-independent effects of previously identified loci was not inadvertently attributed to the mutation-responsive terms for these loci. This process was iterated until no additional *background:locus* terms were discovered for each environment.

**Scans for two-locus effects**. We also performed full-genome scans for two-locus effects. Here every unique pair of loci was interrogated using fixed-effects linear models like those described in the preceding section. As with the one-locus effects, we employed two models in parallel. The model *phenotype ~ background + locus1 + locus2 + locus1:locus2 + error* was used to identify mutation-independent two-locus effects, whereas the model *phenotype ~ background + locus1 + locus2 + background:locus1 + background:locus2 + locus1:locus2 + background:locus1: locus2 + error* was employed to detect mutation-responsive two-locus effects. Specifically, we tested for significance of the *locus1:locus2* and *background:locus1: locus2* interaction terms with the former and latter models, respectively. Simpler terms were included in each model to ensure that variance was not erroneously attributed to more complex terms. Significance thresholds for these terms were determined using 1000 permutations with the correspondence between genotypes and phenotypes randomly shuffled each time. However, to reduce computational run time, 10,000 random pairs of loci, rather than all possible pairs of loci, were examined in each permutation. Significance thresholds were again established based on the fifth quantile of minimum $p$ values observed across the permutations. To ensure that our main findings were robust to threshold, we also generated results at false discovery rates (FDRs) of 0.01, 0.05, and 0.1 by comparing the rate of discoveries at a given $p$ value in the permutations to the rate of discoveries at that same $p$ value in our results (Supplementary Table 10; Supplementary Note 2).

**Scans for three-locus genetic effects**. Owing to computational limitations, we were unable to run a comprehensive scan for mutation-independent and mutation-responsive three-locus effects. Instead, we scanned for three-locus effects involving two loci that had already been identified in a given environment (*known_locus1* and *known_locus2*) and a third locus that had yet to be detected (*locus3*). The model *phenotype ~ background + known_locus1 + known_locus2 + locus3 + known_locus1:known_locus2 + known_locus1:locus3 + known_locus2:locus3 + known_locus1:known_locus2:locus3 + error* was used to identify mutation-independent three-locus effects, whereas the model *phenotype ~ background + known_locus1 + known_locus2 + locus3 + background:known_locus1 + background:known_locus2 + background:locus3 + known_locus1:known_locus2 + known_locus1:locus3 + known_locus2:locus3 + background:known_locus1:known_locus2 + background:known_locus1:locus3 + background:known_locus2:locus3 + known_locus1:known_locus2:locus3 + background:known_locus1:known_locus2: locus3 + error* was employed to detect mutation-responsive three-locus effects. Significance of the *known_locus1:known_locus2:locus3* and *background:known_locus1:known_locus2:locus3* terms in the respective models was determined using 1000 permutations with the correspondence between genotypes and phenotypes randomly shuffled each time. For each permutation, 10,000 trios of sites were chosen by first randomly picking two loci on different chromosomes and then randomly selecting an additional 10,000 sites. The minimum $p$ value across the 10,000 tests was retained. Significance thresholds were again established based on

the fifth quantile of minimum $p$ values observed across the permutations. As with the two-locus effect scans, we also performed our analysis across multiple FDR thresholds to ensure that our findings were robust (Supplementary Table 10; Supplementary Note 2).

**Assignment of mutation-responsive effects to knockouts**. In the aforementioned linkage scans, genetic effects exhibited statistical interactions with the *background* term if they had a different phenotypic effect in at least one of the eight backgrounds relative to the rest. To determine the specific knockouts that interacted with each mutation-responsive effect, we used the contrast() function from the R package lsmeans. This was applied to the specific effect of interest post hoc using the same linear models that were employed for detection. All possible pairwise contrasts between wild-type and knockout segregants were conducted. Mutation-responsive effects were assigned to specific mutations if the contrast between a mutation and a WT population was nominally significant. Unless otherwise noted, we counted each assignment of a mutation-responsive effect to a specific knockout as a separate genetic effect even if they involved the same set of loci.

**Statistical power analysis**. To determine the statistical power of our mapping procedures, we simulated phenotypes for the 1411 genotyped segregants given their genotypes at randomly chosen loci and then tried to detect these loci using the approaches described earlier. In each simulation, a given segregant's phenotype was determined based on both the mutation it carried (if any), as well as its genotype at one, two, or three randomly chosen loci. The effects of mutations were calculated based on the real phenotype data for the glucose environment. Phenotypic effects of the mutation-responsive locus or loci were also attributed to each segregant. Specifically, the phenotype of segregants in only one of the possible genotype classes were increased by a given increment, which we refer to as the absolute effect size. For one-, two-, and three-locus effects, this respectively entailed half, one quarter, and one eighth of the individuals having their phenotypes increased by the increment. In the case of mutation-independent effects, these increments were applied to all eight of the wild-type and knockout backgrounds. In contrast, for mutation-responsive effects, increments were only applied to one of the eight backgrounds, with the specific background randomly chosen. Lastly, random environmental noise was added to each segregant's phenotype. Using these genotype and phenotype data, we tested whether we could detect the loci that had been given a phenotypic effect. This was done by fitting the appropriate fixed-effects linear model, extracting the $p$ value for the relevant term, and determining if that $p$ value fell below a nominal significance threshold of $\alpha = 0.05$. Statistical power was calculated as the proportion of tests at a given phenotypic increment where $p \leq 0.05$. The results of this analysis are shown in Supplementary Fig. 7.

**Contributions of loci involved in higher-order epistasis**. For all mutation-responsive two- and three-locus effects, we determined the proportion of mutation-responsive phenotypic variance explained by each individual locus and the interactions among these loci. To do this, we generated seven subsets of data, each of which were comprised of the wild-type segregants and one set of knockout segregants. We then fit the same model that was used to originally identify a given mutation-responsive effect to the appropriate data subsets. For two-locus effects, we obtained the sum of squares for the *background:locus1*, *background:locus2*, and *background:locus1:locus2* terms. We then divided each of these values by the sum of all three sum of squares. For three-locus effects, we obtained the sum of sum of squares associated with each individual locus (*background:locus1*, *background: locus2*, *background:locus3*) and pair of loci (*background:locus1:locus2*, *background: locus1:locus3*, *background:locus2:locus2*), as well as the sum of squares associated with the trio of loci (*background:locus1:locus2:locus3*). We then divided the total sum of squares associated with each class of terms by the total sum of squares across all mutation-responsive terms. The ternary plots used to show these results were generated using the R package ggtern.

**Analysis of mutation-responsive effects across environments**. We determined whether each one-, two-, and three-locus mutation-responsive effect exhibited a phenotypic effect in any environment outside the one in which it was originally detected. To do this, we used seven subsets of data, each of which was comprised of the wild-type segregants and one set of knockout segregants. We then fit the same model that was used to originally identify a given mutation-responsive effect to the appropriate data subsets for each of the nine additional environments. The $p$ value was then extracted for the relevant term. Bonferroni corrections were used to account for multiple testing.

**Genetic explanation of changes in phenotypic variance**. We measured the phenotypic variance explained by each mutation-responsive genetic effect in the relevant knockout background(s), as well as in the wild-type background. Here we fit each mutation-responsive genetic effect in both populations without using any *background* term. For mutation-responsive one-, two-, and three-locus effects, the following models were respectively employed: *phenotype ~ locus1 + error*, *phenotype ~ locus1 + locus2 + locus1:locus2 + error*, and *phenotype ~ locus1 + locus2 + locus3 + locus1:locus2 + locus1:locus3 + locus2:locus3 + locus1:locus2:locus3 +*

*error*. Partial $R^2$ values were obtained in each population by obtaining the sum of squares associated with the term of interest and dividing it by the total sum of squares. Mutation-responsive effects were then classified by the number of knockout backgrounds in which they were detected. For each class, the number of genetic effects with larger partial $R^2$ values in the knockout background than in the wild-type background (enhanced effects) and the number of genetic effects with smaller partial $R^2$ values in the knockout background than in the wild-type background (reduced effects) were determined. The proportion of mutation-responsive effects that show enhanced and reduced phenotypic effects was calculated for each class. 95% bootstrap confidence intervals were then generated using 1000 random samplings of the data with replacement.

**Checking potential consequences of allele frequency bias.** Allele frequency bias may result in the erroneous detection of mutation-responsive genetic effects due to uneven representation of one-, two-, or three-locus combinations across the knockout and wild-type backgrounds. To account for this, we generated $2 \times 8$, $4 \times 8$, and $8 \times 8$ contingency tables for all one-, two-, or three-locus interactions, respectively, counting each of the possible allele combinations in the wild-type and seven knockout populations. Specifically, for one-locus interactions, we counted the number of individuals carrying the BY and 3S allele at the significant locus for each population. For two-locus interactions, the number of individuals carrying the BY/BY, BY/3S, 3S/BY, and 3S/3S alleles at the two loci were enumerated. For three-locus interactions, the number of individuals carrying the BY/BY/BY, BY/BY/3S, BY/3S/BY, 3S/BY/BY, BY/3S/3S, 3S/BY/3S, 3S/3S/BY, and 3S/3S/3S alleles at the three loci were counted. We then ran chi-square tests to identify individual loci or combinations of loci that show different frequencies across the eight backgrounds, using Bonferroni corrections to account for multiple testing. After filtering out genetic effects that involve loci or combinations of loci with biased frequencies, we repeated our main analyses to ensure that our results were robust to allele frequency differences (Supplementary Figs. 5 and 6; Supplementary Table 11 and 12; Supplementary Note 3).

## Data availability

Genotype and phenotype data, as well as information on all identified loci, are provided in the Supplementary materials.

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

## Acknowledgements

We thank Norm Arnheim, Mark Chaisson, Matt Dean, Sasha Levy, David Pfennig, and Kevin Roy for comments on a draft of this manuscript. We also thank Alessandro Coradini, Jonathan Lee, and Fabian Seidl for input during the execution of this project and writing of this paper. The research described in this manuscript was supported by grant R01GM110255 from the National Institutes of Health, as well as a Computational and Evolutionary Molecular Biology fellowship from the Alfred P. Sloan Foundation to I. M.E. and a Research Enhancement Fellowship from the USC Graduate School to M.N.M. Many of the analyses described in this paper were performed on the USC High-Performance Computing cluster.

## Author contributions

M.N.M., T.M., R.S. and I.M.E. conceptualized this project. M.N.M., T.M., R.S. and R.F. performed experiments. M.N.M., T.M. and I.M.E. analyzed data. M.N.M., T.M., R.S. and I.M.E. wrote the paper.

## Additional information

**Competing interests:** The authors declare no competing interests.

