## [Peer Review File · Nature Communications]

Reviewer #1 (Remarks to the Author):

Mullis et al. address an important question: to what extent (and in what ways) are mutations' effects modified by genetic background? Answers to this question have major implications for understanding the importance of contingency in evolution and for understanding the genetic basis of human disease. As the authors correctly state, genetic background effects have been known for a while, but systematic approaches to characterizing them (and therefore general insights) have been lacking. The authors' study is therefore timely and this line of research has the potential to advance the field substantially.

The approach taken by the authors is to do quantitative-trait locus mapping to uncover background interactions with seven focal mutations (and interactions between the background loci), for yeast growth in 10 environments (different carbon sources or temperatures or chemical challenges). They performed an impressive amount of strain construction and genotyping (>1000 segregants total from eight crosses), as well as an impressive amount of phenotyping (quantitative growth measures of those segregants in each environment). The seven focal mutations are deletions of genes involved in chromatin regulation and transcription, and the focus on this class of genes is well justified by the authors.

This data set and its analysis will ultimately make an important contribution to the literature on epistasis and genetic-background effects. However, there are shortcomings in the way the analysis was conducted that deserve attention.

Major comments:

1) The manuscript is short on detail about how the 47 original genes were pruned down to the 7 focal genes that were used in this study. The rationale of the screen is brief and buried in the legend to SFig 2. The rationale and design of the screen should be made more clear in the main text. What is clear in the current manuscript is that the authors sensibly aimed to pick mutations that would have mappable background effects, but then they need to be careful about how they draw general conclusions (because they are already enriching for interactions).

2) To a large extent the manuscript presents quantitative conclusions about mutation-responsive genetic effects (e.g., numbers of 1-way, 2-way and 3-way interactions with the focal mutation). These quantitative conclusions depend on setting an arbitrary significance threshold. The authors should present an analysis of how robust their findings are to the setting of the significance threshold.

3) The description of the mapping procedure makes it sound as if each of the 30,000+ segregating SNP loci was treated independently. But the single round of meiosis in each cross will produce linked haplotype blocks. It is not clear how this linkage was dealt with in the analysis. There is a Methods section on "Genome-wide coverage of mutation-responsive loci" that mentions reducing to chromosomal intervals, but that appears to be only in the context of determining the fraction of the genome covered by loci that respond to each focal mutation (i.e., not part of the mapping per se). It seems that reducing the effective number of loci by collapsing adjacent loci would increase mapping power in this analysis. On that note, it is not clear how much power the 3-locus vs. 2-locus vs. 1-locus interaction scans have relative to each other or in absolute terms. It is therefore unclear how to think about the relative numbers of discovered interactions of each type. (That said, even if power is low, the authors find a lot of interactions including higher-order ones, which is a very important conclusion of this work.)

4) The authors use the extent to which the same loci interact with different focal mutations as a measure of pleiotropy. There are two issues here. First, the distinction between linkage and pleiotropy is important and has been discussed in the literature quite a bit, so the analysis should somehow address this issue. Second, an opportunity is missed to investigate a more natural measure of pleiotropy: the extent to which the same loci affect growth in different environments. In general, the environmental dimension of the data is glossed over throughout the manuscript. Effects are summed across environments with little analysis of overlaps between environments. The environmental dimension is one of the strengths of the experimental design and should be much more central to the analysis.

Less-major comments:

1) On pg. 8 the authors state that "to our knowledge, our study is the first to explicitly show with genetic mapping data that disruption of a particular gene changes the phenotypic effects of loci on a genome-wide scale". One could argue that the authors' ref. 22 (cited in that sentence) did this for Hsp90, so a different wording might be advisable.

2) On pg. 8 the authors say "Why this knockout modifies the phenotypic effects of so many loci is difficult to infer from current understanding of Hos3's biochemical activities." Is there any relevant information from large-scale genetic interaction data (i.e., Costanzo et al 2016, doi.org/10.1126/science.aaf1420)?

3) On pg. 5 "3 LOD confidence intervals" might be unclear to a general audience.

4) In Fig 2, it is not clear if it shows only mutation-responsive loci for which there is exactly one focal mutation that they respond to (the color key lists individual focal mutations) or if it shows the same loci more than once (once for each focal mutation that they respond to).

5) It is not clear whether "mutation-independent" and "mutation-responsive" are considered mutually exclusive. The statistical models do not appear to make them mutually exclusive and the text on pg. 14 says "We first identified individual loci that show mutation-independent and/or mutation-responsive phenotypic effects", suggesting they are not mutually exclusive. But in the Results section they appear to be treated as mutually exclusive (e.g., on pg. 5 it is stated that 14% of effects were mutation-independent vs. 86% mutation-dependent). This should be clarified.

6) On pg. 16 the term "novel locus" is confusing. Does it exclude other significant loci?

7) On pg. 5 the term "global genetic modifier" is used. This is a useful concept introduced previously by this group. But it seems that it would be hard to set a cutoff of number of interactions above which a mutation would be considered a global modifier, because there is likely to be a continuous distribution of number of interactions. This is one place where the choice of the seven focal mutations matters. Among these mutations, HOS3 stands out as qualitatively different in the scope of its interactions, but had more (or other) focal mutations been chosen, then perhaps HOS3 would fall somewhere on a spectrum of quantitative differences. It might be worthwhile to flesh out the quantitative definition of global genetic modifier rather than treat it as a distinct category.

Reviewer #2 (Remarks to the Author):

In this manuscript, Mullis et al. performed a massive genetic mapping of natural modifiers of the effect of de novo gene knock-outs in yeast. The knock-outs targeted genes coding for chromatin modifiers. The authors studied the phenotype (growth rates in 10 environments) of hundreds of segregants from a cross between two genetically diverse strains (BY and S3) in a context where the de novo KO segregates together with standing natural variation. For each environmental conditions, they used a linear model to infer epistasis between the KO and natural polymorphisms, testing for one-, two- or three- modifying loci. They found thousands of genetic modifying effects (loci of standing variation interacting with the KO in the control of growth rate). They describe the complexity as well as the specificity/pleiotropy of the corresponding effects. They reach the

interesting conclusion that high-order epistasis rather than single-locus effects are responsible for background-mediated modification of the effect of the deletion.

The question addressed (genetic architecture of background effects) is fundamental to genetics and evolutionary biology in general (structure of epistasis). It is also very important to yeast genetics: gene deletions are widely used for mechanistic studies and revealing their complex background-dependency will catch the attention of the community working on chromatin regulations. Although the study is impressive in quantity, I have two major concerns and three suggestions to improve the manuscript.

Major concerns

1. The authors generated gene deletions by transforming the KanMX cassette with gene-flanking homology into the hybrid BY/S3 diploid. My concern is that this diploid already contains two other 'MX' cassettes (HphMX and NatMX) in the genome. In this situation, the majority of KanR transformants that are obtained usually result from recombination between the cassettes (in TEF promoter and terminator) and not between the (shorter) homology sequence designed in the primers to target the gene of interest. The authors mention in methods that they verified deletions by PCR and the primers are indicated in Supp Table. However, for genes having a coding sequence close to 1.4Kb (e.g. CTK1, ESA1, GCN5, EAF1), amplicons from wild-type are expected to have a similar size as amplicons from deletion mutants. Because of the particular context of the selection cassettes, and because the entire paper relies on the quality of the knock-outs, it is important to provide the details of the verification (gel scans or sequencing) and to verify that the diploid remained HygR and NatR after introducing the mutation.

2. A number of conclusions are based on the assignment of responsive effects to specific knock-outs. If I understand correctly page 17, assignment to knock-out X is made by subsetting the data to X and WT only, by applying the same linear models as described pages 14-16, by interrogating the p-value of the responsive effect, and by Bonferroni-correcting this p-value (division by 7, number of KO subsettings). No permutation test is mentioned at this step. The methods should indicate if (as I assume) only the loci detected in the genome scans are considered here. If this is the case, interpretation of the p-value should be made cautiously: a locus I may have a strong modifier effect on KO A, and a very small (marginally significant, not passing the genome scan) on KO B. In this case, I is submitted to the assignment test when searching for modifiers of B (because it figures in the overall list), a small p-value is observed (because of the marginal effect), and I ends up in the responsive effects assigned to B (as would have thousands of marginally-significant loci that did not have the chance to modify A). This would be a wrong assignment, because I modifies A and not B. It is possible that I misunderstood the methods, but if the test was made this way, the authors should find a more rigorous assignment criteria. I can think of post-hoc analysis (e.g. Tukey or Scheffe) on the global linear models instead of subsetting. This would interrogate groups and determine which

mutants are grouped with the WT. There may be other possibilities. This point can have important consequences on the specificity/pleiotropy conclusions of the authors and on Figs 1b, 3, 4a, 4b and 5.

Suggestions for improvement:

3. The authors focused on genetic mapping. They describe KO versus WT traits in the context of recombinants (segregants) but not parental genomes. The paper would gain high value if, for at least a few KO and environments, traits were described in the parental context. Is background effect substantial between BY and S3? Do modifier loci act in the expected direction? Do they explain most of the parental background effects? Since the paper is about explaining background effects (by thousands of QTLs), parental data is important.

4. The results section starts very abruptly. Clarity would be improved if Figure S2 was included in main figure 1, with most of its legend in the main text. Also, the selection of mutants was about $\text{varKO} > \text{varWT}$, but no figure shows directly the comparison of these variances. Dot plots with $y = \text{varKO}$ and $x = \text{varWT}$ should be added.

5. Why not including the environment as a factor in the model (instead of applying one model per environment)? This would allow to quantify (KOxG)xE interactions.

Reviewer #3 (Remarks to the Author):

In this paper, Mullis et al use the budding yeast to systematically explore the genetic ‘architecture’ of background effects. Specifically, the team mapped QTLs in ~200 individual segregants from a wild-type (BY) X 322134S (3S) strain cross, in 7 different genetic backgrounds for 10 growth phenotypes or conditions. They detected “genetic effects”, which include single locus, two loci and 3 loci interactions that significantly contribute to the phenotypic variance observed, in each of the knockout and WT populations. The idea is that if the knockout or genetic background (in this case, different chromatin regulators) has no effect on the phenotype, the same QTLs will be mapped in each background. A significant genetic background effect would result in mapping of different sets of QTLs. Indeed, the authors observed significant differences between genetic backgrounds, with 14% of all genetic effects being “mutation-independent” (mapped in both knockout and WT) vs. 86%

“mutation-specific”. The take-home message is that the genetic background can significantly shape the genetic architecture of a given trait.

This is an interesting idea, and an important problem in genetics. As it stands, however, I do have some major concerns with the study. First, controls for the allele frequencies for the “genetic effects” in the WT and the knockout mapping populations appear to be missing. Specifically, for each single locus, the allele frequency from either parent would be 50%-50%; however, given the low number of segregant pools and the genetic divergence between the parents, it is statistically impossible to get the same representations of two and three loci combinations in the different pools. Note that this could potentially explain the general trend of decreasing numbers of 2 and 3 loci interactions in the “mutation-independent” category vs. the increasing number of 2 and 3 loci interactions in the “mutation-specific” set, as shown in Figure 1. Imagine that you have a lower chance of getting the same higher order combinations in the WT and Knockout pools - this would result in the exact trends observed here.

Second, given the potential allele frequency differences, one cannot be certain that the phenotypic variances observed have the same weight in WT and knockout populations. As shown in Figure 5, there seems to be a correlation between the number of “mutation-specific” and the difference of phenotypic variance knockout/WT. The authors interpreted this as “our results not only explain the genetic architecture of background effects, but also illustrate how interactions between mutations and mutation-responsive loci can influence heritable phenotypic variation within a population.” However, in my opinion, this might as well be an artifact due to differences in allele frequencies. Indeed, there does appear to be a problem with the allele frequencies, even for a single locus (Figure S4a), presumably due to epistasis that is phenotypically relevant to the fitness of the knockout.

For their conclusions to be valid, I think the authors should address the allele frequency issue by separating the analyses by single locus effects from multi-loci effects. They should include the knock out as a genetic variant and map single locus QTLs in knockout, WT and combined populations, which would give a more solid foundation to interpret the background effect.

Reviewer #4 (Remarks to the Author):

I thought this was an interesting paper. My main concern is the lack of biological insight-- there are many statistical results, but no model supported or even proposed that could explain them. For example, why does Hos3 have so many interactions? Why are most mutation-responsive loci involved in 3-way interactions, while mutation-independent loci are not? Why do mutation-specific effects increase effects in mutants, but pleiotropic effects do not? Without any model, the results don't give much insight into what's going on-- as if a car mechanic told you exactly what sounds your car was making and how the sounds change when he removes particular parts, but without diagnosing the underlying issue.

Other questions:

1. KO segregants will have a constant region of no recombination around the KO gene, due to selection for KanMX-- so this could affect the spectrum of interactions seen. Is there any way to tell how much of an issue this is?
2. Related to Fig 5, does average fitness of each KO (among the KO segregants) help explain the number of mutation-specific loci?
3. Why was 3S chosen as a parental strain? Where did it come from?
4. The data set would allow for nice comparisons of GxG vs GxE. For example, do patterns of GxE interactions for KOs resemble the GxG ones reported here? Or are they unrelated?

Reviewers' comments:

Reviewer 1 (Remarks to the Author):

Major comments:

1) The manuscript is short on detail about how the 47 original genes were pruned down to the 7 focal genes that were used in this study. The rationale of the screen is brief and buried in the legend to SFig 2. The rationale and design of the screen should be made more clear in the main text.

We added more details about the screen into the Results and Discussion section, as well as the Supplement.

What is clear in the current manuscript is that the authors sensibly aimed to pick mutations that would have mappable background effects, but then they need to be careful about how they draw general conclusions (because they are already enriching for interactions).

We appreciate recognition that our approach was sensible. We agree with Reviewer 1 and have included sentences more sentences in the text about how the seven genes we focused on were identified through a screen. We have tried not to overstate the findings that come from the existing data.

2) To a large extent the manuscript presents quantitative conclusions about mutation-responsive genetic effects (e.g., numbers of 1-way, 2-way and 3-way interactions with the focal mutation). These quantitative conclusions depend on setting an arbitrary significance threshold. The authors should present an analysis of how robust their findings are to the setting of the significance threshold.

Reviewer 1 is correct that significance thresholds can have a big impact on the results and interpretations of genetic mapping studies, especially those focused on genetic interactions. To address this possibility, we reiterated our work across a number of False Discovery Rate (FDR) thresholds. Although choice of threshold impacts the number of genetic effects that are detected, all of the major results remain the same regardless of threshold. This implies that our main conclusions are robust to threshold. These analyses across thresholds are reported in the Supplement.

3) The description of the mapping procedure makes it sound as if each of the 30,000+ segregating SNP loci was treated independently. But the single round of meiosis in each cross will produce linked haplotype blocks. It is not clear how this linkage was dealt with in the analysis. There is a Methods section on "Genome-wide coverage of mutation-responsive loci" that mentions reducing to chromosomal intervals, but that appears to be only in the context of determining the fraction of the genome covered by loci that respond to each focal mutation (i.e., not part of the mapping per se). It seems that reducing the effective number of loci by collapsing adjacent loci would increase mapping power in this analysis.

Among the 36,675 high-confidence markers that were genotyped in the cross, only 8,311 unique allele configurations were observed across segregants. This is because often there were no recombination events between closely linked markers. All of our genetic mapping analyses were performed on a set of 8,311 markers that were representative of the unique allele configurations, rather than on the complete set of 36,675 markers. This is now described in the Methods.

On that note, it is not clear how much power the 3-locus vs. 2-locus vs. 1-locus interaction scans have relative to each other or in absolute terms. It is therefore unclear how to think about the relative numbers of discovered interactions of each type. (That said, even if power is low, the authors find a lot of interactions including higher-order ones, which is a very important conclusion of this work.)

To address this point, we performed statistical power analyses. To maximize the relevance of these power analyses, we used our existing genotype data and then simulated phenotypes for each segregant. In these simulations, a given segregant's phenotype was determined based on both the mutation it carried (if any), as well as its genotype at one, two, or three randomly chosen loci. The effects of mutations were calculated based on the real phenotype data. Then, the phenotypic effects of the mutation-responsive locus or loci were added. Lastly, random environmental noise was added to each segregant's phenotype. Using these genotype and phenotype data, we tested whether we could detect the loci that had been given a phenotypic effect, employing a nominal significance threshold of $\alpha = 0.05$. The results of this analysis are in the Supplement and are consistent with Reviewer's 1 expectation. At most effect sizes, we had highest, medium, and lowest power to detect interactions between a mutation and one, two, and three loci, respectively.

4) The authors use the extent to which the same loci interact with different focal mutations as a measure of pleiotropy. There are two issues here. First, the distinction between linkage and pleiotropy is important and has been discussed in the literature quite a bit, so the analysis should somehow address this issue.

Reviewer 1 is right that the issue of linkage and pleiotropy should be discussed. For this reason, we now explicitly mention this point in the Results and Discussion. However, it is not clear how much can be done to analytically resolve this issue because, on average, loci detected in our study spanned around 10 genes. Also, we have eliminated usage of the word 'pleiotropy' because it is a word that is used in different ways in different research areas and we would like to minimize confusion.

Second, an opportunity is missed to investigate a more natural measure of pleiotropy: the extent to which the same loci affect growth in different environments. In general, the environmental dimension of the data is glossed over throughout the manuscript. Effects are summed across environments with little analysis of overlaps between environments. The environmental dimension is one of the strengths of the experimental design and should be much more central to the analysis.

It is standard in the complex traits genetics community, especially the yeast part of this community, to treat measurements from different environments as different traits during genetic mapping. With this said, we tried to address Reviewer 1's point by determining whether mutation-responsive effects impact phenotype in any environments outside the one in which they were originally detected. This analysis is included in the resubmission. In brief, we find that most of mutation-responsive effects are not detectable in other environments. These analyses across environments suggest that the loci involved in background effects typically interact not only with each other and a mutation, but also the environment.

Less-major comments:

1) On pg. 8 the authors state that "to our knowledge, our study is the first to explicitly show with genetic mapping data that disruption of a particular gene changes the phenotypic effects of loci on

a genome-wide scale". One could argue that the authors' ref. 22 (cited in that sentence) did this for Hsp90, so a different wording might be advisable.

We eliminated this text.

2) On pg. 8 the authors say "Why this knockout modifies the phenotypic effects of so many loci is difficult to infer from current understanding of Hos3's biochemical activities." Is there any relevant information from large-scale genetic interaction data (i.e., Costanzo et al 2016, doi.org/10.1126/science.aaf1420)?

Unfortunately, the Costanzo data provide little insight into Hos3. Multiple studies about Hos3 were published in the last year or two, and we have tried to update our pointers to these papers. However, better understanding Hos3's role in interacting with standing variation will require additional studies in the future.

3) On pg. 5 "3 LOD confidence intervals" might be unclear to a general audience.

We eliminated this text.

4) In Fig 2, it is not clear if it shows only mutation-responsive loci for which there is exactly one focal mutation that they respond to (the color key lists individual focal mutations) or if it shows the same loci more than once (once for each focal mutation that they respond to).

We changed Figure 2 so that it is no longer color-coded by knockout populations to reduce confusion. We show the same loci more than once if they interact with multiple mutations. We modified the legend for this figure to make this work easier to understand.

5) It is not clear whether "mutation-independent" and "mutation-responsive" are considered mutually exclusive. The statistical models do not appear to make them mutually exclusive and the text on pg. 14 says "We first identified individual loci that show mutation-independent and/or mutation-responsive phenotypic effects", suggesting they are not mutually exclusive. But in the Results section they appear to be treated as mutually exclusive (e.g., on pg. 5 it is stated that 14% of effects were mutation-independent vs. 86% mutation-dependent). This should be clarified.

Mutation-independent and mutation-responsive loci are considered mutually exclusive. The models used to detect mutation-responsive loci include potential mutation-independent effects of involved loci because not doing this could create false positives. This would occur if phenotypic variance explained by mutation-independent effects loaded onto the mutation-dependent terms. This is now explained in more detail in the Methods.

6) On pg. 16 the term "novel locus" is confusing. Does it exclude other significant loci?

We have removed this part of the analysis to minimize confusion. The results of this focused scan for two-locus effects were very similar to the results of the unbiased, genome-wide two-locus scan. Thus, there was little impact of excluding this analysis from the paper.

7) On pg. 5 the term "global genetic modifier" is used. This is a useful concept introduced previously by this group. But it seems that it would be hard to set a cutoff of number of interactions above which a mutation would be considered a global modifier, because there is likely to be a continuous distribution of number of interactions. This is one place where the choice of the seven focal mutations matters. Among these mutations, HOS3 stands out as

qualitatively different in the scope of its interactions, but had more (or other) focal mutations been chosen, then perhaps HOS3 would fall somewhere on a spectrum of quantitative differences. It might be worthwhile to flesh out the quantitative definition of global genetic modifier rather than treat it as a distinct category.

We agree with Reviewer 1's point and eliminated usage of this term.

Reviewer 2 (Remarks to the Author):

Major concerns

1. The authors generated gene deletions by transforming the KanMX cassette with gene-flanking homology into the hybrid BY/S3 diploid. My concern is that this diploid already contains two other 'MX' cassettes (HphMX and NatMX) in the genome. In this situation, the majority of KanR transformants that are obtained usually result from recombination between the cassettes (in TEF promoter and terminator) and not between the (shorter) homology sequence designed in the primers to target the gene of interest. The authors mention in methods that they verified deletions by PCR and the primers are indicated in Supp Table. However, for genes having a coding sequence close to 1.4Kb (e.g. CTK1, ESA1, GCN5, EAF1), amplicons from wild-type are expected to have a similar size as amplicons from deletion mutants. Because of the particular context of the selection cassettes, and because the entire paper relies on the quality of the knock-outs, it is important to provide the details of the verification (gel scans or sequencing) and to verify that the diploid remained HygR and NatR after introducing the mutation.

We added more detail into the Methods and Supplement. A number of checks were done to ensure that a given gene of interest was completely deleted by *KanMX*. First, given that strains carried *HphMX*, *KanMX*, and *NatMX*, we conducted selections for transformants on G418, ClonNAT, and Hygromycin B at the same time. In addition, we used PCR checks to ensure that *KanMX* was integrated into the genome at the expected site and in the expected orientation. Lastly, we checked the whole genome sequencing data for each set of knockout segregants. We obtained reads that contained junctions between *KanMX* and the genome, and checked whether the part of the genome joined to *KanMX* was the expected part of the genome.

2. A number of conclusions are based on the assignment of responsive effects to specific knock-outs. If I understand correctly page 17, assignment to knock-out X is made by subsetting the data to X and WT only, by applying the same linear models as described pages 14-16, by interrogating the p-value of the responsive effect, and by Bonferroni-correcting this p-value (division by 7, number of KO subsetings). No permutation test is mentioned at this step. The methods should indicate if (as I assume) only the loci detected in the genome scans are considered here. If this is the case, interpretation of the p-value should be made cautiously: a locus I may have a strong modifier effect on KO A, and a very small (marginally significant, not passing the genome scan) on KO B. In this case, I is submitted to the assignment test when searching for modifiers of B (because it figures in the overall list), a small p-value is observed (because of the marginal effect), and I ends up in the responsive effects assigned to B (as would have thousands of marginally-significant loci that did not have the chance to modify A). This would be a wrong assignment, because I modifies A and not B. It is possible that I misunderstood the methods, but if the test was made this way, the authors should find a more rigorous assignment criteria. I can think of post-hoc analysis (e.g. Tukey or Scheffe) on the global linear models instead of subsetting. This would interrogate groups and determine which mutants are grouped with the WT. There may be other

possibilities. This point can have important consequences on the specificity/pleiotropy conclusions of the authors and on Figs 1b, 3, 4a, 4b and 5.

To address this issue, we used the “contrast” function from the R package `lsmeans` on the global linear models to test all pairwise contrasts between mutation and WT populations. Mutation-responsive effects were assigned to specific mutations if the contrast between a mutation and a WT population was nominally significant. Using these new assignments, we re-ran all downstream analysis and re-made all figures. In general, we found that most of the conclusions from the paper remain consistent using this new strategy.

Suggestions for improvement:

3. The authors focused on genetic mapping. They describe KO versus WT traits in the context of recombinants (segregants) but not parental genomes. The paper would gain high value if, for at least a few KO and environments, traits were described in the parental context. Is background effect substantial between BY and S3? Do modifier loci act in the expected direction? Do they explain most of the parental background effects? Since the paper is about explaining background effects (by thousands of QTLs), parental data is important.

This would certainly be interesting. However, in attempting this work, we had issues deleting some of the seven focal genes in both BY and 3S. This is not necessarily surprising. For example, *ESAI* is annotated as an essential gene in the *Saccharomyces* Genome Database. It is also well known that genes can vary in their essentiality among yeast strains. Thus, we unfortunately were not able to provide parental data for this manuscript.

4. The results section starts very abruptly. Clarity would be improved if Figure S2 was included in main figure 1, with most of its legend in the main text. Also, the selection of mutants was about $\text{varKO} > \text{varWT}$, but no figure shows directly the comparison of these variances. Dot plots with $y = \text{varKO}$ and $x = \text{varWT}$ should be added.

Figure S2 is now Figure 1. We also now clarify in the Results section how the deletion screen was conducted and how the seven focal genes were chosen. In general, we used Levene’s test to determine knockouts that cause significant changes in phenotypic variation when deleted. However, gene knockouts with lower phenotypic variance may also be significant using Levene’s test if a knockout population contains outliers. To emphasize the presence of these outliers, we changed the x-axis to be coefficient of variation instead of mean. We also highlight the WT population in the figure so that it is easier to directly compare the changes in phenotypic variance and coefficient of variation between WT and knockout populations.

5. Why not including the environment as a factor in the model (instead of applying one model per environment)? This would allow to quantify (KOxG)xE interactions.

Linkage mapping studies in yeast and other organisms typically separately perform genetic mapping on data from different environments. Additionally, the statistical models employed in this study are already nearing the bounds of complexity that can be explored using the current data. That said, we completely agree in the importance of studying KOxGxE and tried to pursue this analysis in a different way, which is described in the Results. As Reviewer 2 will see, our results suggest that the genetic basis of background effects is highly environment-dependent. This result is consistent with our past work (see Lee et al. 2016. PLOS Genetics), which was conducted on a smaller scale.

Reviewer 3 (Remarks to the Author):

This is an interesting idea, and an important problem in genetics. As it stands, however, I do have some major concerns with the study. First, controls for the allele frequencies for the “genetic effects” in the WT and the knockout mapping populations appear to be missing. Specifically, for each single locus, the allele frequency from either parent would be 50%-50%; however, given the low number of segregant pools and the genetic divergence between the parents, it is statistically impossible to get the same representations of two and three loci combinations in the different pools. Note that this could potentially explain the general trend of decreasing numbers of 2 and 3 loci interactions in the “mutation-independent” category vs. the increasing number of 2 and 3 loci interactions in the “mutation-specific” set, as shown in Figure 1. Imagine that you have a lower chance of getting the same higher order combinations in the WT and Knockout pools - this would result in the exact trends observed here.

In the paper, we show that mutation-independent effects tend to be genetically simpler, while mutation-responsive effects tend to be more genetically complex. Reviewer 3 proposes this might be a technical artifact driven by allele frequency differences between the different knockout and wild type versions of the BYx3S cross. To examine this possibility, we excluded loci that show biased individual or multi-locus allele frequencies from our analyses, and then conducted our main analyses again. When we did this, our core findings remain the same, implying that our main findings are not driven by technical artifacts associated with allele frequencies. This analysis is described in the Supplementary materials.

Second, given the potential allele frequency differences, one cannot be certain that the phenotypic variances observed have the same weight in WT and knockout populations. As shown in Figure 5, there seems to be a correlation between the number of “mutation-specific” and the difference of phenotypic variance knockout/WT. The authors interpreted this as “our results not only explain the genetic architecture of background effects, but also illustrate how interactions between mutations and mutation-responsive loci can influence heritable phenotypic variation within a population.” However, in my opinion, this might as well be an artifact due to differences in allele frequencies. Indeed, there does appear to be a problem with the allele frequencies, even for a single locus (Figure S4a), presumably due to epistasis that is phenotypically relevant to the fitness of the knockout.

To control for potential allele frequency effects in this analysis, we analyzed the relationship between phenotypic variance and detected loci for each knockout population individually, using data from all 10 environments. This analysis is robust to allele frequency differences because allele frequencies are constant within a given knockout version of the BYx3S cross. When we do this, we find nominally significant correlations between identified mutation-responsive effects and changes in phenotypic variance in all seven knockout populations. We used permutations to examine the probability of observing this result by chance. Across 100,000 permutations, we never observed more than three knockout versions of the BYx3S cross showing correlations by chance, suggesting that our result is highly significant. This work is described in the Results.

For their conclusions to be valid, I think the authors should address the allele frequency issue by separating the analyses by single locus effects from multi-loci effects. They should include the knock out as a genetic variant and map single locus QTLs in knockout, WT and combined populations, which would give a more solid foundation to interpret the background effect.

We chose to use the combined segregant data to maximize statistical power. We addressed the allele frequency problems using methods described above. Also, we would argue that many of the downstream analyses strongly support the idea that the identified loci have biologically meaningful effects.

Reviewer 4 (Remarks to the Author):

I thought this was an interesting paper. My main concern is the lack of biological insight-- there are many statistical results, but no model supported or even proposed that could explain them. For example, why does Hos3 have so many interactions? Why are most mutation-responsive loci involved in 3-way interactions, while mutation-independent loci are not? Why do mutation-specific effects increase effects in mutants, but pleiotropic effects do not? Without any model, the results don't give much insight into what's going on-- as if a car mechanic told you exactly what sounds your car was making and how the sounds change when he removes particular parts, but without diagnosing the underlying issue.

We appreciate Reviewer 4's positive view and acknowledge that their following comments represent important future directions. This study aims to provide insight into the structure and frequency of background effects, rather than provide detailed mechanisms underlying the cellular response to each knockout. Better understanding the biology requires a more comprehensive study of background effects in response to a wider range of knockouts and/or determining the polymorphisms underlying genetic effects via cloning.

Other questions:

1. KO segregants will have a constant region of no recombination around the KO gene, due to selection for KanMX-- so this could affect the spectrum of interactions seen. Is there any way to tell how much of an issue this is?

Reduced recombination and allelic bias around the knockout cassettes has the potential to cause both false negatives and false positives. To deal with this problem, we excluded the 50 kb regions upstream and downstream of the knockouts from our main analyses. Also, as shown in the Supplementary materials and described earlier in this response, we redid our main analyses after filtering sites that show allele or multi-locus genotype frequency bias. These biased sites include regions flanking knockouts. When we exclude biased sites, our central findings remain qualitatively unchanged.

2. Related to Fig 5, does average fitness of each KO (among the KO segregants) help explain the number of mutation-specific loci?

This an interesting idea, but no, average fitness of each knockout among the segregants does not explain the number of mutation-specific loci.

3. Why was 3S chosen as a parental strain? Where did it come from?

The 322134S strain is a clinical isolate from New Castle, UK. It was chosen for this study because we have previously used this cross to study background effects involving a yeast colony phenotype (see Taylor et al. 2014. PLOS Genetics; Taylor et al. 2015. PLOS Genetics; Taylor et al. 2016. Nature Communications; Lee et al. 2016. PLOS Genetics). The current study represents

an effort to expand the scale and generality of our ongoing work on background effects. For this reason, it is natural to work on the same cross as our previous research.

4. The data set would allow for nice comparisons of GxG vs GxE. For example, do patterns of GxE interactions for KOs resemble the GxG ones reported here? Or are they unrelated?

Please see response to Reviewer 1, major comment 4.

Reviewer #1 (Remarks to the Author):

The authors have done an excellent job of addressing concerns about the original version of the manuscript. My own concerns have been addressed with new analyses and revision of the text. The other reviewers also voiced valid concerns about the original manuscript, which the authors have taken seriously and addressed through new analyses and text revisions. For example, Reviewer 3's concerns about allele frequencies were addressed by re-running analyses after excluding loci with biased allele frequencies and by conducting correlation analyses across environments separately for the different crosses (within which allele frequencies are fixed). The current manuscript is much improved over the original manuscript and presents a timely, well-conducted study of broad interest.

Reviewer #2 (Remarks to the Author):

Regarding concern 1, the revised text now satisfyingly explains how the authors verified the gene deletions made in the parental diploid.

Regarding concern 2, the authors now use a post-hoc analysis with the `lsmeans::contrast` function to assign responsive effects to specific knock-outs. This is appropriate and the issue has been addressed in Fig 5a and c. However, I am now totally confused by the y-axis of Fig 6 (and S9-S10), which is different from Fig 5 of the initial submission. What is this y-axis exactly? "enhanced effects" and "reduced effects" are not defined in legend, nor in main text, nor in methods:

- I initially assumed that an "enhanced effect" corresponded to a background:locus interaction having the same sign as the background term in the model; and a "reduced effect" corresponded to opposite signs. If this is correct, then the interest of Fig 6 should be better explained: it is not surprising that enhancement of effects correlates with $V_{mut} > V_{wt}$, but enhancement can take place via a strong magnitude of the effects (value of the coefficients of interaction term) or via polygenicity (small individual magnitude of each effect, but numerous effects). It seems to me that Fig6. supports the latter scenario: the number of effects matters. If this is correct, then showing the distributions of coefficient values (magnitudes of effects) would be useful.

- Alternatively, the authors may have simply called "enhanced effects" the effects where $PVE_{ko} > PVE_{wt}$. In this case, if more loci have $PVE_{ko} > PVE_{wt}$ (high y value), it is not surprising that $V_{mut} > V_{wt}$ (high x value), unless few loci are responsible for high V_{mut} . Again, the authors should explain if/how Fig6 allows to distinguish between polygenicity and magnitude.

Minor revisions remaining:

- line 182: It is necessary to mention the post-hoc analysis here and not only in methods. Otherwise, readers won't understand what was done.

- line 190: Fig5c not Fig 5b.

- line 207: Fig6, not Fig 5.

- line 666: remind the definition of PVE in legend (even if defined in Fig3 legend)

Reviewer #4 (Remarks to the Author):

The authors have addressed most of my concerns, except for the lack of biological insight.

Line 171 has a typo, "mutation-respects"

June 25, 2018

To the Editors and Reviewers:

We appreciate the time you have taken to review our paper. We attempted to address your comments in this second resubmission. Please see below for a point-by-point response.

Sincerely,

Martin Mullis, Takeshi Matsui, and Ian Ehrenreich

Reviewer 1:

1. The authors have done an excellent job of addressing concerns about the original version of the manuscript. My own concerns have been addressed with new analyses and revision of the text. The other reviewers also voiced valid concerns about the original manuscript, which the authors have taken seriously and addressed through new analyses and text revisions. For example, Reviewer 3's concerns about allele frequencies were addressed by re-running analyses after excluding loci with biased allele frequencies and by conducting correlation analyses across environments separately for the different crosses (within which allele frequencies are fixed). The current manuscript is much improved over the original manuscript and presents a timely, well-conducted study of broad interest.

Thank you for your positive words regarding the manuscript.

Reviewer #2:

1. Regarding concern 1, the revised text now satisfyingly explains how the authors verified the gene deletions made in the parental diploid.

We are glad that this point has been addressed in a satisfactory manner. It was an important point.

2. Regarding concern 2, the authors now use a post-hoc analysis with the `lsmeans::contrast` function to assign responsive effects to specific knock-outs. This is appropriate and the issue has been addressed in Fig 5a and c. However, I am now totally confused by the y-axis of Fig 6 (and S9-S10), which is different from Fig 5 of the initial submission. What is this y-axis exactly? "enhanced effects" and "reduced effects" are not defined in legend, nor in main text, nor in methods:

- I initially assumed that an "enhanced effect" corresponded to a background:locus interaction having the same sign as the background term in the model; and a "reduced effect" corresponded to opposite signs. If this is correct, then the interest of Fig 6 should be better explained: it is not surprising that enhancement of effects correlates with $V_{mut} > V_{wt}$, but enhancement can take place via a strong magnitude of the effects (value of the coefficients of interaction term) or via polygenicity (small individual magnitude of each effect, but numerous effects). It seems to me that Fig6. supports the latter scenario: the number of effects matters. If this is correct, then showing the distributions of coefficient values (magnitudes of effects) would be useful.

- Alternatively, the authors may have simply called "enhanced effects" the effects where $PVE_{ko} > PVE_{wt}$. In this case, if more loci have $PVE_{ko} > PVE_{wt}$ (high y value), it is not surprising that $V_{mut} > V_{wt}$ (high x value), unless few loci are responsible for high V_{mut} . Again, the authors

should explain if/how Fig6 allows to distinguish between polygenicity and magnitude.

These constructive comments are very helpful. To address them, we now explicitly define “enhanced” and “reduced” in the main text. These terms are defined based on phenotypic variance explained (‘PVE’) in the mutants relative to the wild type segregants. We have tried to make this much clearer in multiple places. We have also provided a new Supplementary Figure focused on the differences in PVE by each mutation-responsive effect in mutants and wild type segregants. We also now state that the pattern in Figure 6 arises due to polygenicity. Although the pattern in Figure 6 is not necessarily surprising given our genetic mapping results, it is an important figure to show given the work of people like Susan Lindquist and protégés, as well as Mark Siegal, who have emphasized how perturbation of Hsp90 and potentially other genes can alter the heritable phenotypic variation within populations. We added a sentence to help place this figure and the associated analysis in its broader intellectual context.

3. line 182: It is necessary to mention the post-hoc analysis here and not only in methods. Otherwise, readers won't understand what was done.

We have tried to make this clearer by adding two sentences to the main text.

4. line 190: Fig5c not Fig 5b.

We made this modification.

5. line 207: Fig6, not Fig 5.

We made this modification.

6. line 666: remind the definition of PVE in legend (even if defined in Fig3 legend)

We made this modification.

Reviewer #4:

1. The authors have addressed most of my concerns, except for the lack of biological insight.

We are glad that we addressed most of your concerns.

2. Line 171 has a typo, "mutation-respects"

We made this modification.

Reviewer #2 (Remarks to the Author):

Fig6 y-axis is now satisfyingly explained and I agree with the authors' interpretation of this figure.